# Genome-Wide Identification of Solute Carrier Family 12 and Functional Characterization of Its Role in Saline–Alkaline Stress Acclimation in the Ridgetail White Shrimp *Exopalaemon carinicauda*

**DOI:** 10.3390/ijms26178339

**Published:** 2025-08-28

**Authors:** Shuai Tang, Jiajia Wang, Kuo Yan, Zhixin Yu, Jitao Li

**Affiliations:** 1State Key Laboratory of Mariculture Biobreeding and Sustainable Goods, Yellow Sea Fisheries Research Institute, Chinese Academy of Fishery Sciences, Qingdao 266071, China; 18837625451@163.com (S.T.); wangjj@ysfri.ac.cn (J.W.); yankuo87@163.com (K.Y.); 19863931069@163.com (Z.Y.); 2Function Laboratory for Marine Fisheries Science and Food Production Processes, Qingdao Marine Science and Technology Center, Qingdao 266237, China; 3National Experimental Teaching Demonstration Center of Aquatic Science, Shanghai Ocean University, Shanghai 201306, China

**Keywords:** solute carrier family 12, transmembrane ion transport, osmotic regulation, saline–alkaline acclimation

## Abstract

Solute carrier family 12 (*SLC12*) encodes electroneutral cation-coupled chloride cotransporters responsible for transmembrane ion transport (Na^+^, K^+^, and Cl^−^), which play a critical role in aquatic osmoregulation. However, the *SLC12* gene of *Exopalaemon carinicauda* (*EcSLC12*) has not been systematically identified or functionally characterized. In this study, six *EcSLC12* genes were identified across the genome and classified into N(K)CC, KCC, CCC9, and CIP subfamilies. Three *NKCC1* homologous genes (*EcSLC12A2.1*, *EcSLC12A2.2*, and *EcSLC12A2.3*) were reported for the first time in crustaceans. The *EcSLC12* family exhibited distinct expression patterns in response to low-salinity, high-alkalinity, and saline–alkaline stress. *EcSLC12A2.2* was highly expressed in the gill, and its expression was closely correlated with saline–alkaline acclimation. Additionally, *EcSLC12A2.2* knockdown decreased *E. carinicauda* survival under saline–alkaline stress. Thus, *EcSLC12A2.2* plays critical roles in osmotic regulation and saline–alkaline acclimation. This study provides crucial insights into *E. carinicauda*’s saline–alkaline tolerance mechanisms, and the discovery of multiple *NKCC1* homologs fills a gap in the crustacean *SLC12* gene family research.

## 1. Introduction

In China, there exists 1.6 × 10^7^ hm^2^ of saline–alkaline land and 7.6 × 10^6^ hm^2^ of low-lying saline–alkaline water [1]. Saline–alkaline water is generally defined as water with a salinity exceeding 1 ppt and carbonate alkalinity exceeding 3 mmol/L [2]. Compared to freshwater and seawater, saline–alkaline water is characterized by increased carbonate alkalinity, high pH levels, and imbalanced ionic composition [3]. These conditions impose physiological stress on aquatic animals by causing ionic dysregulation, osmoregulatory failure, and histopathological damage [4,5,6]. Despite their vast resources, saline–alkaline waters remain largely underutilized. Developing aquaculture in these environments by selecting tolerant aquatic species presents a promising strategy for ecosystem restoration and the expansion of aquaculture resources [7]. Aquatic animals that can tolerate such an environment depend on efficient ion transport mechanisms to maintain osmotic homeostasis. For marine species, the lower salinity of saline–alkaline water (compared to seawater) means they need enhanced active uptake of key ions, such as Na^+^ and Cl^−^, to sustain osmotic balance [8,9]. In contrast, freshwater species exposed to saline stress adapt by increasing the excretion of excess ions, including Na^+^ and HCO_3_^−^ [10,11]. For example, *Paramisgurnus dabryanus* upregulates carrier-mediated ion and ammonia transport under high alkalinity to maintain osmotic equilibrium and alleviate impaired ammonia excretion [12]. Similarly, in *Eriocheir sinensis*, the lncRNA–miRNA–mRNA regulatory axis enhances aerobic metabolism to support ion transport enzymes, thereby strengthening ion regulation and osmotic stability [13]. However, the specific ion transport mechanisms that enable aquatic crustaceans to acclimate to saline–alkaline environments remain insufficiently understood and warrant further investigation.

Solute carriers (SLCs) are the largest transmembrane transporter superfamily capable of transporting sugars, amino acids, nucleotides, ions, and drugs, and they play a crucial role in maintaining cellular and systemic homeostasis [14,15]. The members of the solute carrier family 12 (*SLC12*) encode cation-chloride cotransporters (CCCs) that facilitate the movement of Na^+^, K^+^, and Cl^−^ across membranes [16]. In vertebrates, at least nine members (*SLC12A1-SLC12A9*) have been identified [17,18,19]. This family is characterized by the presence of an amino acid permease domain (AA_permease), a solute carrier family 12 domain (SLC12A), and multiple transmembrane regions within the central hydrophobic region [20]. The *SLC12* family plays a crucial role in osmoregulation and salinity acclimation in aquatic animals. *SLC12A2* (*NKCC1*) and *SLC12A1* (*NKCC2*) are abundantly expressed in the epithelial cells of osmoregulatory tissues, such as the gills and intestines, in teleost fishes. These cotransporters interact with Na^+^/K^+^-ATPase (NKA) and Cl^−^ channels to regulate ion transport [21,22]. In crustaceans, *NKCC1* expression is modulated in response to changes in environmental salinity. For instance, elevated salinity induces transient upregulation of *NKCC1* expression in the gills of *Litopenaeus vannamei* and the antennal glands of *Scylla paramamosain*. Moreover, studies in palaemonid shrimp indicate that the NKCC protein facilitates ion uptake in freshwater but mediates Cl^−^ efflux under high-salinity environments [23,24,25]. These findings highlight the essential role of the SLC12 family in osmoregulation across diverse aquatic species.

*Exopalaemon carinicauda* is a commercially important shrimp species in the Yellow Sea and Bohai Sea regions of China [26]. This species displays remarkable adaptability to saline–alkaline environments, maintaining normal reproductive capacity at a salinity of 2 ppt and achieving sustainable growth at carbonate alkalinities of up to 8 mmol/L [27,28]. *E*. *carinicauda* has been cultivated on a commercial scale in some saline–alkaline areas (approximate carbonate alkalinity: 3.5–13 mmol/L) [27]. Analysis of the saline–alkaline adaptability of *E*. *carinicauda* has crucial theoretical and practical significance for developing new tolerant varieties and improving the utilization efficiency of saline–alkaline waters.

A comprehensive genome-wide annotation of the *SLC12* family has not been performed, despite its emerging role in crustacean osmoregulation, and functional characterization remains limited to individual genes. Herein, we report the first systematic identification of the *SLC12* family in *E. carinicauda*, combining a genome-wide approach with expression profiling in response to saline–alkaline stress and RNAi-mediated functional validation of *SLC12A2*. This study provides insights into the evolutionary diversification of the *SLC12* genes and their potential applications in aquaculture under saline–alkaline environments.

## 2. Results

### 2.1. Identification of the EcSLC12 Gene Family

Six *EcSLC12* genes were identified in the *E. carinicauda* genome. These genes were designated as *EcSLC12A2.1*, *EcSLC12A2.2*, *EcSLC12A2.3*, *EcSLC12A6*, *EcSLC12A8*, and *EcSLC12A9*, following standard nomenclature protocols for human *SLC12* genes.

Table 1 shows the physicochemical analysis of the EcSLC12 proteins and the prediction of their subcellular localization. The length of the EcSLC12 proteins ranged from 638 to 1057 aa, the molecular weight ranged from 68.76 kDa to 117.19 kDa, and the theoretical isoelectric point ranged from 5.68 to 8.21. The instability index ranged from 35.08 to 42.72, and the aliphatic index ranged from 98.7 to 104.11. The grand average hydrophobicity ranged from 0.091 to 0.262. The total average hydrophobic index of all EcSLC12 proteins was >0, and the predicted hydrophilicity plot of EcSLC12 proteins is shown in Appendix A. Subcellular localization predictions indicated that all EcSLC12 proteins were membrane-localized, except for the EcSLC12A2.3 protein (endoplasmic reticulum). Additionally, transmembrane domain prediction revealed that all proteins possessed 8–13 hypothetical transmembrane domains (Figure 1).

### 2.2. Gene Structures, Conserved Motifs, and Domains of the EcSLC12 Gene Family

The structural analysis of *EcSLC12* genes revealed conserved patterns associated with evolutionary relationships (Figure 2). Some differences were observed in the number of exons, conserved motifs, and domains among the members of the *EcSLC12* gene family. The exon number varied from 9 to 20; most *EcSLC12* genes had 20 exons, and *EcSLC12A9* and *EcSLC12A8* had fewer exons (17 and 9, respectively) (Figure 2a). Conserved motifs were shared among closely related genes: *EcSLC12A2.1* and *EcSLC12A2.2* had 15 motifs; *EcSLC12A2.3* had 14 motifs (lost motif 14); and *EcSLC12A6*, *EcSLC12A9*, and *EcSLC12A8* had 10, 9, and 6 motifs, respectively (Figure 2b). All EcSLC12 proteins had the “AA_permease” domain, and all proteins except for EcSLC12A8 also had the “SLC12A” domain (Figure 2c).

### 2.3. Chromosome Location and Phylogenetics of the EcSLC12 Gene Family

A distribution map of the *EcSLC12* genes in *E. carinicauda* chromosomes was generated based on the GFF file (Figure 3). This analysis showed that six *EcSLC12* genes were distributed across four chromosomes and one scaffold. *EcSLC12A9*, *EcSLC12A6*, and *EcSLC12A8* were localized on chromosomes 12, 39, and 42, respectively, whereas *EcSLC12A2.1* and *EcSLC12A2.2* were positioned on chromosome 28. Additionally, *EcSLC12A2.3* was mapped to scaffold512, which was not assembled onto the chromosome in a previous study [29].

A phylogenetic tree was constructed using SLC12 protein sequences from *E. carinicauda* and other representative species to determine the phylogenetic relationships within the SLC12 gene family (Figure 4). The 80 protein sequences were divided into the Na^+^-coupled chloride cotransporter group (NKCC and NCC), K^+^, Cl^−^ cotransporter group (KCC), cation-chloride-cotransporter 9 group (CCC9), and cotransporter interacting protein group (CIP), according to the constructed phylogenetic relationships. *EcSLC12A2.1–2.3* belong to the NKCC group. Notably, *EcSLC12A2.1* and *EcSLC12A2.2* cluster together, while *EcSLC12A2.3* groups with *SLC12A2* from *Macrobrachium rosenbergii*. This suggests that the duplication event giving rise to the branches containing *EcSLC12A2.1/2.2* and *EcSLC12A2.3* occurred earlier, and this was then followed by a further duplication that produced the *EcSLC12A2.1* and *EcSLC12A2.2*. Additionally, *EcSLC12A6* and *EcSLC12A9* are closely related to those in *Penaeus japonicus*. Genes in the KCC and CIP groups share a common ancestral branch, indicating closer evolutionary relationships.

Additionally, *SLC12* genes of vertebrates from the same group clustered together, suggesting that the *SLC12* family underwent differentiation between vertebrates and invertebrates during evolution.

### 2.4. Expression of the EcSLC12 Genes in Various Tissues

The expression patterns of *EcSLC12* were systematically analyzed in the eight tissues of *E. carinicauda* (Figure 5). *EcSLC12A2.1* and *EcSLC12A2.2* demonstrated significantly higher expression in the gill than in the other tissues (*p* < 0.05). *EcSLC12A2.3*, *EcSLC12a6*, and *EcSLC12A9* showed significantly higher expression in the hepatopancreas than in the other tissues (*p* < 0.05), followed by the intestine and blood, with minimal expression in the gills. In contrast, *EcSLC12A8* expression was higher in the muscle than in the other tissues (*p* < 0.05).

### 2.5. Analysis of the Expression Pattern of EcSLC12 Genes Under Varying Stress Conditions

Expression changes of *EcSLC12* genes in the gills were analyzed after *E. carinicauda* was transferred from seawater to low-salinity, high-alkalinity, and saline–alkaline conditions (Figure 6). After low-salinity stress, *EcSLC12A2.2* expression sharply increased, peaking at 12 h with a 90-fold increase compared to the control group (*p* < 0.05). In contrast, *EcSLC12A2.1* was significantly downregulated throughout the 6–72 h stress period (*p* < 0.05). *EcSLC12A2.3* and *EcSLC12A9* revealed transient suppression within the first 6 h, followed by gradual recovery to the initial levels. After high-alkalinity stress, *EcSLC12A2.2* expression peaked at 3 h, followed by a gradual decline and a drop below the control group levels at 12 h (*p* < 0.05). *EcSLC12A8* expression peaked at 6 h and recovered to its initial level at 72 h. *EcSLC12A2.1*, *EcSLC12A6*, and *EcSLC12A9* were predominantly downregulated throughout the stress period, with transient upregulation noted at 24 and 48 h. After saline–alkaline stress, *EcSLC12A2.2* expression peaked at 6 h and decreased to the initial level at 72 h. In contrast, the gene expressions of *EcSLC12A2.1*, *EcSLC12A2.3*, *EcSLC12A6*, and *EcSLC12A9* were inhibited.

### 2.6. Saline–Alkaline Stress Analysis After EcSLC12A2.2 Silencing

Pilot experiments demonstrated that dsRNA-a induced stable and efficient gene silencing for up to 72 h post-injection under unstressed conditions (Figure 7a) and was therefore used in subsequent stress experiments. In the saline–alkaline stress experiment (Figure 7b), *EcSLC12A2.2* expression was significantly reduced at 24, 48, and 72 h after RNAi treatment, with expression levels consistently lower in the dsRNA-a group than in controls (*p* < 0.05). Knockdown efficiency ranged from 53% to 81%, confirming effective silencing of *EcSLC12A2.2*. Concomitant with transcriptional silencing, shrimp survival rates demonstrated a progressive decline in the RNAi group at 46.6% at 72 h and 40% at 96 h after interference; both values were significantly lower than those in the control group (*p* < 0.05; Figure 7c). Taken together, these results suggest that *EcSLC12A2.2* plays a crucial role in the acclimation of *E. carinicauda* to fluctuations in salinity and alkalinity.

## 3. Discussion

### 3.1. Identification and Evolutionary Analysis of the SLC12 Family in E. carinicauda

This study is the first to identify the *SLC12* family in the *E. carinicauda* genome, providing crucial insights into ion transport mechanisms in this crustacean. The *SLC12* family has been extensively characterized in vertebrates, yet its role in crustaceans remains limited [30,31]. Here, six *EcSLC12* genes were identified, whereas five classic *SLC12* genes (*SLC12A1*, *SLC12A3*, *SLC12A4*, *SLC12A5*, and *SLC12A7*) were absent. *SLC12A1* (*NKCC2*) and *SLC12A3* (*NCC*) are widely conserved in vertebrates and known to mediate ion transport in teleosts [32,33,34]. Outside of vertebrates, *SLC12A3* homologs have been identified in mollusks, such as *Sinonovacula constricta* [35]. However, there is currently no genomic evidence supporting the presence of either *SLC12A1* or *SLC12A3* in crustaceans. The *SLC12A4–SLC12A7* subfamily corresponds to the potassium-chloride cotransporters (KCCs). Phylogenetic analysis reveals that in vertebrates, *SLC12A4* and *SLC12A6* form distinct monophyletic clades, indicating independent evolutionary origins. For example, all human *KCC* genes (*SLC12A4*–*SLC12A7*) have undergone independent evolutionary divergence and generate functionally diverse splice variants through alternative splicing [36]. In contrast, only a single *KCC* gene (*SLC12A4/A6*) has been identified in certain crustaceans, including *E. carinicauda*. Moreover, putative *SLC12A4* and *SLC12A6* homologs from the same crustacean species cluster closely on the phylogenetic tree and share high sequence similarity (Appendix A), indicating that they likely represent alternative splice variants of a single gene.

The *SLC12* family is less diverse in invertebrates than in vertebrates [37,38]. This phylogenetic disparity is likely due to whole-genome duplication (WGD) events, which drive the rapid expansion of gene families and subsequent neofunctionalization or subfunctionalization of paralogs [39,40]. Notably, vertebrates likely underwent two early WGD events that significantly contributed to vertebrate genome expansion and complexity [41,42]. Consequently, the absence of specific *SLC12* genes in *E. carinicauda* and other invertebrates may reflect the lack of these vertebrate-specific WGD events. Phylogenetic analysis revealed that *EcSLC12* genes cluster with those of crustaceans but are distinct from those of mammalian and teleost fish. This evolutionary divergence between vertebrate and invertebrate lineages is consistent with findings from other studies on the *SLC12* family [43,44].

All EcSLC12 proteins contain the “AA_permease” domain, a structural feature conserved in the SLC12 counterparts across species [38]. Proteins with this domain are frequently associated with transmembrane transport of amino acids, such as proteins of the amino acid-polyamine-organo cation (APC) superfamily [45]. Additionally, most EcSLC12 proteins contain the “SLC12A” domain at the C-terminus, a domain critical for the transport of K^+^ and Cl^−^ ions [46]. EcSLC12A8 lacks the “SLC12A” domain, a feature also observed in 14 other species (Appendix A). Analysis of protein motifs shows that motifs 1, 2, 4, 7, 9, 12, and 14 are located in the “AA_permease” domain, whereas motifs 3, 5, 6, 8, and 11 are located in the “SLC12A” domain. These conserved motifs likely help to maintain the structural stability and functional activity of EcSLC12 proteins.

### 3.2. Functional Study of the EcSLC12 Family in Acclimating to Saline–Alkaline Environments

The identification of three *NKCC1* genes (*EcSLC12A2.1*, *EcSLC12A2.2*, and *EcSLC12A2.3*) in *E. carinicauda* represents the first reported case of *NKCC1* gene expansion in crustaceans. This suggests that gene duplication has been instrumental in the adaptive evolution of osmoregulatory mechanisms in this group. The *SLC12* family is known for its considerable evolutionary plasticity. In teleost fishes, the *NKCC* gene subfamily contains two primary members, *NKCC1* and *NKCC2*, and, in many species, both have undergone duplication, generating paralogous pairs (e.g., *NKCC1a*/*NKCC1b* and *NKCC2α*/*NKCC2β*) [47,48]. These duplicated genes exhibit divergent tissue-specific expression patterns and are likely to have evolved distinct physiological roles. This diversification represents an important adaptive mechanism enabling teleosts to inhabit environments with varying salinities. However, the current scarcity of crustacean genomic resources hinders a comprehensive understanding of how prevalent *NKCC1* duplication is within this group. Further comparative genomic studies are required to determine whether this expansion is unique to *E. carinicauda* or widespread among crustaceans.

In aquatic organisms, gills not only facilitate respiration but also act as the primary interface between the hemolymph and the external environment, contributing to ion regulation and nitrogenous waste excretion [49]. Given these multifunctional roles, gills are crucial for maintaining osmotic balance under fluctuating environmental conditions. In *E. carinicauda*, gills are essential for acclimation to saline–alkaline stress [50]. Consistent with this, *EcSLC12A2.1* and *EcSLC12A2.2* exhibit the highest expression levels in gill tissue, suggesting critical roles in osmoregulation. Typically, the NKCC1 protein is localized to the basolateral membrane of ionocytes, where it mediates the uptake of ions like Na^+^, K^+^, and Cl^−^ from the hemolymph. These ions are subsequently utilized intracellularly or transported to the external environment via other ion transporters or channels. During freshwater acclimation, *NKCC1* gene expression is generally downregulated, which is a pattern also observed for *EcSLC12A2.1* in *E. carinicauda* [51,52,53]. In contrast, *EcSLC12A2.2* is significantly upregulated after transfer from high to low salinity conditions, suggesting functional divergence between these homologs.

Interestingly, the expression pattern of *EcSLC12A2.2* under varying salinity conditions resembles that of the sodium-chloride cotransporter (*NCC*) in fish. In fish, gill ionocytes are classified into seawater-type (SW) and freshwater-type (FW) cells based on the localization and function of ion transporters [54,55]. NCC/NCC2, encoded by *SLC12A3* or *SLC12A10*, is apically localized and mediates Na^+^/Cl^−^ cotransport in a 1:1 ratio [56]. Its expression is strongly upregulated during freshwater acclimation in various fish species [57,58]. Similar to other crustaceans, *E. carinicauda* reduces water influx and increases ion uptake during low-salinity adaptation [59,60]. Therefore, it is plausible that the ion transporter encoded by *EcSLC12A2.2* may be localized to the apical membrane of gill ionocytes, facilitating environmental ion entry into the cell. Despite this functional resemblance, direct functional evidence for *NCC* or its homologs in crustaceans is currently lacking. Phylogenetic analysis places *EcSLC12A2.2* within the *NKCC1* subfamily, which is evolutionarily distant from vertebrate *NCCs*. Therefore, the hypothesis that *EcSLC12A2.2* has converged functionally with *NCCs* requires further verification

In the gills of *E. carinicauda*, the ion transporters encoded by *EcSLC12A2.1* and *EcSLC12A2.2* are likely expressed in distinct subpopulations of ionocytes, indicating functional specialization. The expansion of *NKCC1* genes in this species may represent an independent evolutionary acclimation to increased demands for ion transport, potentially signifying a unique innovation within crustaceans. Notably, *EcSLC12A2.2* rapidly responds to high-alkalinity stress and contributes to the regulation of ion balance. This response may be triggered by elevated Na^+^ concentrations in water resulting from NaHCO_3_ supplementation, which temporarily activates *EcSLC12A2.2* and facilitates extracellular Na^+^ influx. However, under prolonged alkaline stress, the activity of *EcSLC12A2.2* is significantly suppressed, promoting active ion excretion and reducing the uptake of external ions. Functionally, *EcSLC12A2.2* plays an essential role in both salinity acclimation and alkalinity stress resistance. This is further supported by RNA interference experiments, in which knockdown of *EcSLC12A2.2* resulted in significantly increased mortality under combined saline–alkaline stress, confirming its critical role in maintaining ion homeostasis and organism survival under environmental challenge.

Although the artificial saline–alkaline water used in this study exceeded the alkalinity typically observed in natural environments [61], *E*. *carinicauda* demonstrated exceptional tolerance to both low salinity and extreme alkalinity. The dynamic regulation of the *EcSLC12* gene family members is central to this osmoregulatory ability. These findings not only clarify the molecular mechanisms of stress tolerance in this species but also offer valuable insights for enhancing resilience in aquaculture systems.

## 4. Materials and Methods

### 4.1. Identification and Bioinformatic Analysis of the SLC12 Family in E. carinicauda

#### 4.1.1. Database Resources

The genome sequence, gene annotation, and protein sequences of *E. carinicauda* were obtained from whole genome sequencing results [29], and other species (*Homo sapiens*, *Mus musculus*, *Danio rerio*, *Oreochromis niloticus*, *Dicentrarchus labrax*, *Drosophila melanogaster*, *Cherax quadricarinatus*, *Macrobrachium rosenbergii*, *Homarus americanus*, *Penaeus japonicus*, *Macrobrachium nipponense*, *Penaeus chinensis*, *Portunus trituberculatus*, and *Scylla paramamosain*) were obtained from the National Center for Biotechnology Information (https://www.ncbi.nlm.nih.gov/, accessed on 23 May 2024).

#### 4.1.2. Identification and Sequence Analysis of *SLC12* Family in *E. carinicauda*

Initially, HMM profiles corresponding to the *SLC12* family (PF03522 and PF00324) were obtained from the InterPro database (https://www.ebi.ac.uk/interpro/entry/pfam/, accessed on 27 May 2024) and used to screen the protein sequences of *E. carinicauda* for potential *EcSLC12* candidates. To minimize potential false negatives, *SLC12* genes from *H. sapiens*, *D. rerio*, and *D. melanogaster* were used as references to identify potential *EcSLC12* genes in the *E. carinicauda* genome through TBLASTN and BLASTP searches. Candidate sequences with E-values < 1 × 10^−5^ and alignment coverage >70% were retained for further analysis.

The physicochemical properties of the identified proteins—including amino acid length, molecular weight (MW), isoelectric point (pI), instability index, aliphatic index, grand average of hydropathicity (GRAVY), and hydrophobicity—were predicted using the online tool Expasy-ProtScale (https://web.expasy.org/protscale/, accessed on 10 June 2024). Subcellular localizations were predicted using Cell-PLoc 2.0 (http://www.csbio.sjtu.edu.cn/bioinf/Cell-PLoc-2/, accessed on 10 June 2024), and transmembrane domains were inferred with TMHMM-2.0 (https://services.healthtech.dtu.dk/services/TMHMM-2.0/, accessed on 12 June 2024).

#### 4.1.3. Phylogenetic Analysis of the *SLC12* Family

The EcSLC12 protein sequences were integrated with homologous proteins from 15 species. A multi-species homologous sequence alignment was performed using MEGA11 software, and a phylogenetic tree was constructed using the maximum likelihood method (model: LG+G+F, bootstrap replicates: 1000). The phylogenetic tree was visualized and refined using iTOL (Version 7.1) (https://itol.embl.de/itol.cgi, accessed on 5 October 2024).

#### 4.1.4. Gene Structure Analysis, Conserved Motif/Domain, and Chromosomal Localization Identification

The gene structures and chromosomal positions of *EcSLC12* genes were determined by parsing the genome annotation file (GFF3 format) using TBtools. Conserved protein motifs were predicted via the MEME Suite (v5.5.7, https://meme-suite.org/meme/tools/meme, accessed on 15 June 2024) with 15 motifs specified and default parameters. Domain architectures were identified using the SMART database(https://smart.embl.de/, accessed on 15 June 2024) with an E-value cutoff of 1 × 10^−5^. All results were visualized using TBtools v2.0.

### 4.2. Functional Characterization of the SLC12 Family in E. carinicauda Under Differential Salinity and Alkalinity Conditions

#### 4.2.1. Expression Pattern of the *EcSLC12* Genes in Various Tissues and Stress Conditions

All healthy adult *E. carinicauda* individuals (mean body length: 3.41 ± 0.27 cm; mean body weight: 0.89 ± 0.20 g) used in this study were obtained from a commercial aquaculture farm in Rizhao, China. Before experimentation, the shrimp were acclimated for one week in 200 L polyvinyl chloride (PVC) tanks equipped with aeration and filled with sand-filtered seawater (salinity: 22 ppt; pH: 8.0 ± 0.11; carbonate alkalinity: 2.35 ± 0.12 mmol/L; temperature: 29 ± 1 °C). After acclimatization, approximately 400 individuals were randomly selected for experimental use. The experimental water conditions were established by adding NaHCO_3_ to mixtures of freshwater and seawater to achieve target salinity and alkalinity levels. The experiment was carried out in multiple 200 L aerated PVC tanks, and no water exchange was performed during the experimental period. Carbonate alkalinity was quantified using an acid-titration method [62], and salinity was maintained by adjusting the proportion of freshwater and measured using a salinometer (PAL-06S, ATAGO, Tokyo, Japan).

To analyze the expression levels of the *EcSLC12* genes across various tissues, nine shrimp cultured in seawater were randomly selected and grouped into three pooled samples (three shrimp per sample). Tissues, including the heart, blood (hemolymph), muscle, gill, stomach, eyestalk, hepatopancreas, and intestine, were collected from each group and stored at −80 °C for subsequent RNA extraction.

Three acute stress trials were conducted under the following conditions: low salinity (5.50 ± 0.41 ppt; alkalinity: 2.32 ± 0.11 mmol/L), high alkalinity (salinity: 22.20 ± 0.52 ppt; alkalinity: 13.73 ± 0.29 mmol/L), and saline–alkaline stress (salinity: 5.57 ± 0.50 ppt; alkalinity: 13.91 ± 0.33 mmol/L). Each treatment group consisted of three replicates, with ten shrimp per replicate. Gill tissues were collected from three unstressed shrimp maintained in standard seawater as control samples (0 h). Thereafter, gill samples were taken from three shrimp (one per replicate) at 1, 3, 6, 12, 24, 48, and 72 h after stress exposure, pooled together, and immediately frozen at −80 °C for RNA extraction. Shrimp were not fed during the experiment. This study aimed to examine changes in the expression patterns of the *EcSLC12 genes* under these distinct stress conditions.

#### 4.2.2. RNAi of *EcSLC12A2.2*

*EcSLC12A2.2* was selected for further functional validation through RNA interference (RNAi) due to its expression patterns in various tissues and saline–alkaline stress. Three specific dsRNAs targeting *EcSLC12A2.2* were designed using the online website SnapDragon for designing double-stranded RNA (dsRNA) primer s (https://www.flyrnai.org/cgi-bin/RNAi_find_primers.pl, accessed on 15 September 2024), each incorporating a T7 promoter sequence at the 5′ end. The amplified products were purified using the Gel Extraction Kit (D2500, Omega Bio-Tek, Guangzhou, China), followed by dsRNA synthesis using the T7 RNAi Transcription Kit (TR102). The synthesized dsRNA was subsequently purified and diluted to a final concentration of 100 ng/μL.

The interference efficiency of different dsRNAs was first assessed under standard seawater conditions. Forty shrimp (0.85 ± 0.24 g) were randomly assigned to four groups and injected with different dsRNAs; shrimp injected with dsGFP served as the control group. The most effective dsRNA was then selected for use in the saline–alkaline challenge experiments (salinity: 5.50 ± 0.51 ppt; alkalinity: 19.91 ± 0.32 mmol/L). For the RNAi-mediated knockdown experiment, 60 healthy *E. carinicauda* individuals (0.89 ± 0.20 g) received an intramuscular injection of target-specific dsRNA at a dosage of 1 μg/g body weight. Control shrimp were injected with an equivalent volume of dsGFP. Gill tissues were sampled from three individuals per group (one per replicate) at 24, 48, and 72 h post-injection for total RNA extraction. Knockdown efficiency was evaluated via quantitative real-time PCR (qRT-PCR). To assess the physiological consequences of gene silencing, survival rates under saline–alkaline stress were recorded at 24, 48, 72, and 96 h for each treatment group, with three replicates per group. Death was defined as the complete loss of voluntary movement and absence of response to strong external stimuli.

#### 4.2.3. RNA Extraction and qRT-PCR Analysis

Total RNA was extracted from the heart, eyestalk, gill, hepatopancreas, stomach, muscle, intestine, and blood tissues of *E. carinicauda* using the TransZol Up Plus RNA kit (Trans, Beijing, China) according to the instruction manual. The cDNA was obtained through reverse transcription using HiScript III RT SuperMix for qPCR expression analysis.

The relative expression of *EcSLC12* genes were calculated using the 2^−∆∆CT^ method, and results were expressed as mean ± standard deviation (mean ± SD) [63]. The qPCR amplifications were carried out in triplicate in a total volume of 20 μL containing 10 μL 2 × ChamQ SYBR Color qPCR Master Mix (Low ROX Premixed), 0.8 μL each of forward and reverse primer (10 μM), 2 μL of the cDNA template, and 7.2 μL of DEPC-treated water. The PCR program was 95 °C for 30 s, then 40 cycles of 95 °C for 5 s and 60 °C for 30 s, followed by 1 cycle of 95 °C for 15 s, 60 °C for 60 s, and 95 °C for 15 s. The primers for RT-PCR are shown in Table 2. β-actin was an internal reference gene because of its stable expression [64].

No animals, experimental units, or data points were excluded from the analysis for any of the experimental groups. All data collected were included in the statistical analysis as planned. Before parametric analysis, the assumptions of normality and homogeneity of variances were statistically tested. One-way analysis of variance (ANOVA) was conducted using SPSS 22.0 to evaluate significant differences among groups. If the data violated the assumptions of normality or homogeneity of variance, non-parametric tests were employed. A *p*-value less than 0.05 (*p* < 0.05) was considered statistically significant. Line and column graphs were generated using GraphPad Prism 8.0 based on the experimental data.

## 5. Conclusions

In this study, a comprehensive genomic and functional characterization of the *SLC12* family in *E. carinicauda* was conducted. Through a genome-wide analysis, six *EcSLC12* genes were identified, including three members classified under the *NKCC1* subtype. This is the first identification of *SLC12* family members in crustaceans, with a gene number less than that observed in vertebrates. The expression pattern of *EcSLC12A2.1* closely resembles that of *NKCC1* in other species, suggesting a conserved functional role. In contrast, *EcSLC12A2.2* exhibits an expression pattern similar to that of *NCC* in teleosts. The ion transporter encoded by *EcSLC12A2.2* may be localized to the apical membrane of ionocytes and is responsible for the transport of Na^+^ and Cl^−^ from the external environment into the cell. This hypothesis requires further experimental validation. Finally, *EcSLC12* genes also play a role in alkalinity, making them key factors in *E. carinicauda*’s acclimation to saline–alkaline aquaculture environments.

## Figures and Tables

**Figure 1 ijms-26-08339-f001:**
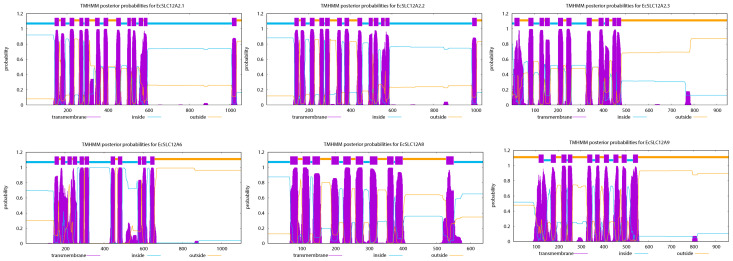
Predicted transmembrane topology of the EcSLC12 protein. The horizontal axis indicates protein length, while the vertical axis shows the prediction probability (or accuracy). The orange, blue, and purple lines correspond to protein regions outside of the membrane, inside of the membrane, and within the transmembrane domain, respectively.

**Figure 2 ijms-26-08339-f002:**
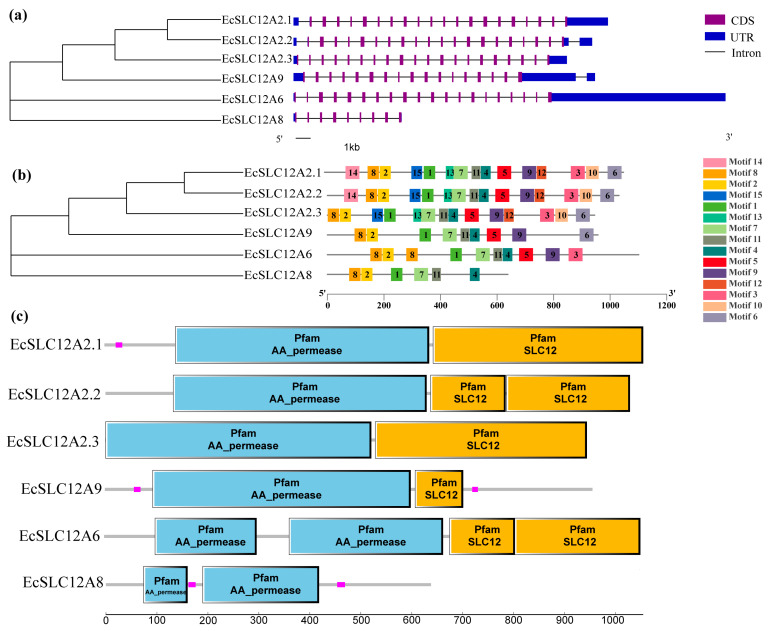
Evolutionary relationships, motif patterns, conserved domains, and gene structure of the EcSLC12 gene family in E. carinicauda. (**a**) The architecture of EcSLC12 genes. The purple box represents the coding sequence; the blue box represents untranslated regions; the black line represents the intron. (**b**) The conserved motif of EcSLC12 proteins. Fifteen presumptive motifs are indicated with boxes marked in different colors. (**c**) Distributions of conserved domains in SLC12 proteins. The blue box represents “AA_permease”, the yellow box represents the “SLC12” domain, and the low-complexity domain is represented in pink.

**Figure 3 ijms-26-08339-f003:**
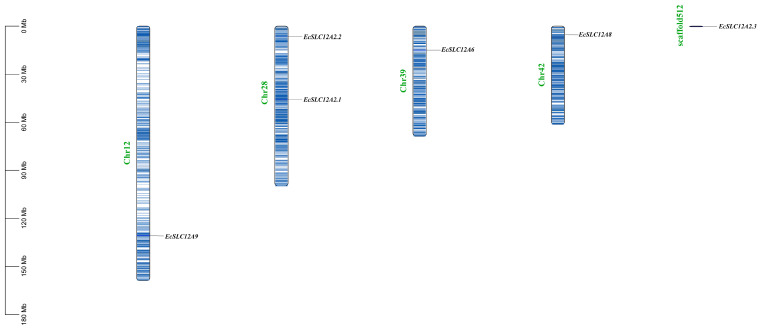
Chromosomal distribution of *EcSLC12* genes in *E. carinicauda*. The scale is used to indicate the length of the chromosome. The bars refer to a total of four chromosomes and one scaffold. The color gradient of chromosomes from white to blue represents gene density from low to high. The names of the chromosomes are written on the left side, and the names of the related genes are written on the right side of the chromosomes.

**Figure 4 ijms-26-08339-f004:**
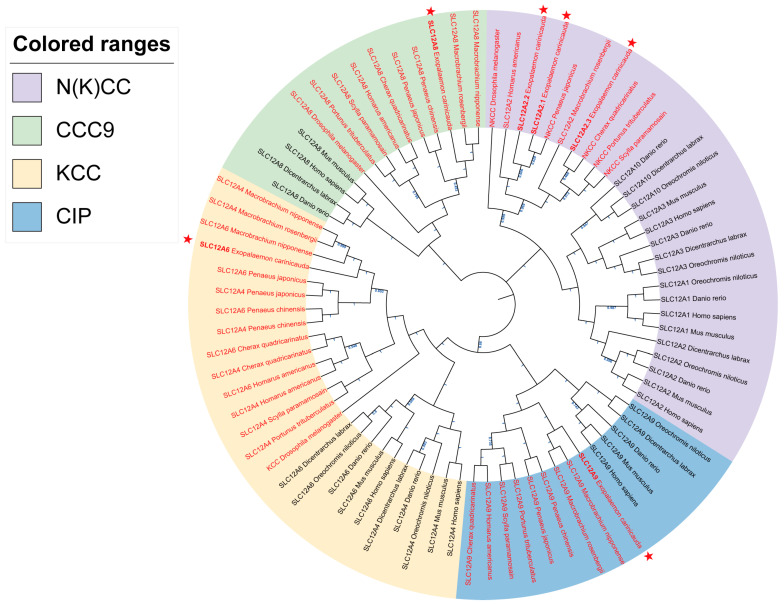
Phylogenetic analysis of *SLC12* genes in *E. carinicauda* and other animals. The tree is built using Mega v11.0 with 1000 bootstrap duplicates based on the maximum likelihood method. The blue font on the branch indicates different levels of bootstrap support, ranging from 0.453 to 1. *EcSLC12* genes are shown in bold and marked with a ★. *SLC12* genes from invertebrates are highlighted in red. The phylogenetic tree is divided into four colored clades: the green region represents the CCC9 group (*SLC12A8*), the blue region the CIP group (*SLC12A9*), the pink region the N(K)CC group (*SLC12A1*, *SLC12A2*, *SLC12A3*, *SLC12A10*, and *NKCC*), and the yellow region the KCC group (*SLC12A4*, *SLC12A6*, and *KCC*). Gene IDs for all *SLC12* are provided in Appendix A.

**Figure 5 ijms-26-08339-f005:**
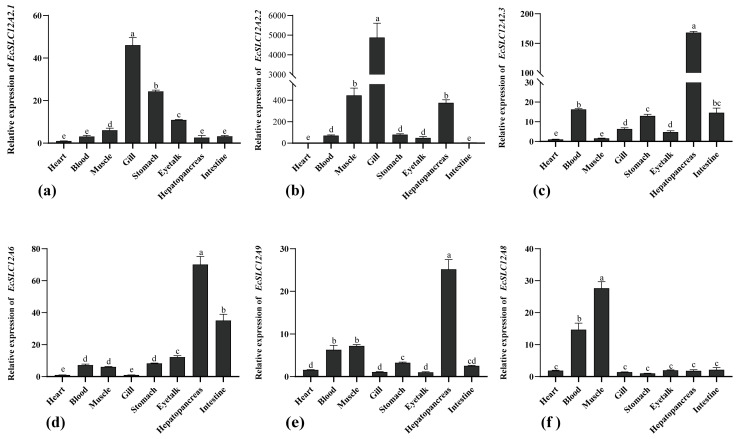
Expression of six *EcSLC12* genes in different tissues of *E. carinicauda*. (**a**): *EcSLC12A2.1*, (**b**): *EcSLC12A2.2*, (**c**): *EcSLC12A2.3*, (**d**): *EcSLC12A6*, (**e**): *EcSLC12A8*, and (**f**): *EcSLC12A9*. Different lowercase letters indicate significant differences in the different tissues (*p* < 0.05). All experiments were performed independently at least three times. Error bars represent the standard deviation of three replicates. The β-actin serves as the internal reference gene.

**Figure 6 ijms-26-08339-f006:**
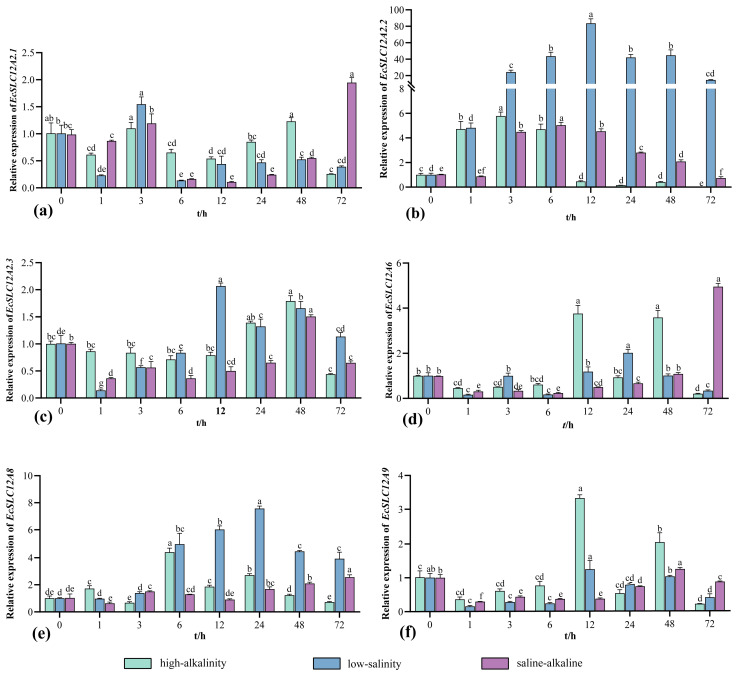
Expression levels of *EcSLC12* genes in the gills of *E. carinicauda* under different stress conditions are shown. (**a**): *EcSLC12A2.1*, (**b**): *EcSLC12A2.2*, (**c**): *EcSLC12A2.3*, (**d**): *EcSLC12A6*, (**e**): *EcSLC12A8*, and (**f**): *EcSLC12A9*. Throughout the experiment, low-salinity conditions were maintained at 5.50 ± 0.41 ppt salinity and 2.32 ± 0.11 mmol/L carbonate alkalinity; high-alkalinity conditions at 22.20 ± 0.52 ppt salinity and 13.73 ± 0.29 mmol/L carbonate alkalinity; and saline–alkaline conditions at 5.57 ± 0.50 ppt salinity and 13.91 ± 0.33 mmol/L carbonate alkalinity. Different lowercase letters indicate significant differences between time points under the same condition (*p* < 0.05). All experiments were performed independently at least three times. Error bars represent the standard deviation of three replicates. β-actin was used as the internal reference gene.

**Figure 7 ijms-26-08339-f007:**
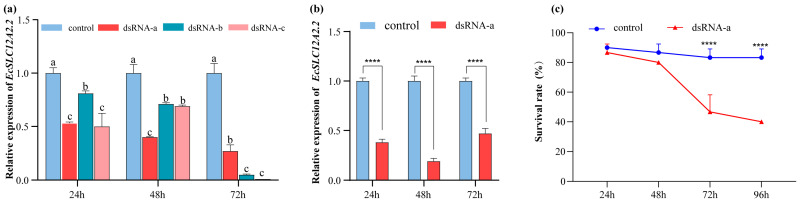
Results of dsRNA interference. (**a**) Efficiency of different dsRNA targets. dsRNA-a, dsRNA-b, and dsRNA-c target the *EcSLC12A2.2* , with dsGFP serving as the control. dsRNA was injected at 1 μg per gram of shrimp body weight. After injection, all shrimp were maintained in seawater. (**b**) Expression levels of *EcSLC12A2.2* following knockdown. The dsGFP-injected group served as the control. Post-injection, shrimp were transferred to saline–alkaline water (salinity: 5.50 ± 0.51 ppt; carbonate alkalinity: 19.91 ± 0.32 mmol/L). (**c**) Survival rate of *E. carinicauda* after *EcSLC12A2.2* knockdown. The dsGFP-injected group served as the control. After injection, all shrimps were placed in saline–alkaline water. Asterisks indicate significant differences between groups. (****: *p* ≤ 0.0001). Different lowercase letters denote significant differences across time points (*p* < 0.05). Experiments were performed independently at least three times. Error bars represent the standard deviation of three replicates. β-actin was used as the internal reference gene.

**Table 1 ijms-26-08339-t001:** Sequence characteristics of EcSLC12 proteins and subcellular location prediction.

Name	aa Number	MW (kDa)	pI	Instability Index	Aliphatic Index	GRAVY	Subcellular Location
EcSLC12A2.1	1057	117.19	5.68	42.72	98.7	0.091	Cell membrane
EcSLC12A2.2	1029	113.51	6.2	41.46	99.52	0.152	Cell membrane
EcSLC12A2.3	945	103.83	5.89	36.44	102.99	0.171	Endoplasmic reticulum
EcSLC12A6	1052	116.58	8.21	35.62	100.67	0.155	Cell membrane
EcSLC12A8	638	68.76	5.98	35.40	98.97	0.202	Cell membrane
EcSLC12A9	957	105.4	6.35	35.08	104.11	0.262	Cell membrane

aa: amino acid; MW: molecular weight; pI: isoelectric point; GRAVY: grand average of hydropathicity.

**Table 2 ijms-26-08339-t002:** Primer design for *EcSLC12* gene expression and RNAi.

Primer	Primer Sequence (5′–3′)	Size (bp)
For qRT-PCR		
EcSLC12A2.2-F	CGTTGCACCGCAAGTTATGG	105
EcSLC12A2.2-R	CAGAAGCGTCAAACCTCCGTCATC	
EcSLC12A2.1-F	TTCACCAGGAAACAACCCAAAGG	84
EcSLC12A2.1-R	CAGGATATAAGGAAGCAGCAGAGTC	
EcSLC12A2.3-F	CGAAGGCAGCAACCAGAACATC	83
EcSLC12A2.3-R	AGAGATTAGCACCCGCAACAATG	
EcSLC12A6-F	TTGCTAAGAACTTGACTGAGGGAATC	110
EcSLC12A6-R	GTGACTGACGCCAACCATACG	
EcSLC12A8-F	GCTCTGGTGGTGACAGTGATG	110
EcSLC12A8-R	CAGGGATTCGCCTTTGG	
EcSLC12A9-F	CACAGGCTTCTCATTGACAACATTG	89
EcSLC12A9-R	GCTTGGTCGTCGTCATCATCTAC	
Ecβ-actin-F	AACTTTCAACACCCCAGCCA	96
Ecβ-actin-R	TCTCCAGAGTCCAGCACGAT	
For RNAi		
dsRNA-a-F	GATCACTAATACGACTCACTATAGGGTGTTGACGTTGAAAACCCAA	300
dsRNA-a-R	GATCACTAATACGACTCACTATAGGGCTTATCCACTGGTGGCTGGT	
dsRNA-b-F	GATCACTAATACGACTCACTATAGGGTTTGCGTTATTGGTATGGCA	486
dsRNA-b-R	GATCACTAATACGACTCACTATAGGGGGAGCAGACACCAAAGAAGC	
dsRNA-c-F	GATCACTAATACGACTCACTATAGGGATGATGAGATTGCCAAAGGC	561
dsRNA-c-R	GATCACTAATACGACTCACTATAGGGGGAGCAGACACCAAAGAAGC	
GFPT7-F	GATCACTAATACGACTCACTATAGGGATGGTGAGCAAGGGGGAGGA	741
GFPT7-R	GATCACTAATACGACTCACTATAGGGTTACTTGTACAGCTCGTCCA	

## Data Availability

Data are contained within the article or the Appendix A.

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
