# Peer review of "Genome-Wide Identification of Solute Carrier Family 12 and Functional Characterization of Its Role in Saline–Alkaline Stress Acclimation in the Ridgetail White Shrimp Exopalaemon carinicauda"

_ijms, 2025, doi:10.3390/ijms26178339_

Round 1
Reviewer 1 Report
Comments and Suggestions for Authors
Comments and observations for the work “Genome-Wide Identification of the Solute Carrier Family 12 and Functional Characterization of Their Roles in Saline-Alkaline Stress Adaptation in Exopalaemon carinicauda”. This work is interesting from the perspective of understanding the saline-alkaline mechanisms and tolerance on E. carinicauda , This can be useful in various species. The results are novel, however, some issues should be attended:
Title: I recommend adding the common name.
Abstract:
The abstract sections are well-defined; however, they substantially add to the methodology used (techniques, processes, analysis). Please note the maximum word count.
Introduction:
L43: Negative ions are represented in various ways in the text; apply the same criteria, e.g., L55.
L46-50: This section can be more specific, providing examples with other species of aquatic organisms or other aquatic life, as well as mentioning the physiological and molecular changes referred to in more detail. This will allow the reader to better understand the text.
L62-64: Same as the previous comment, please be more specific.
L100: Change “,” to “:”
Figure 1: The figure is illegible; it is recommended to improve the resolution and increase the size. The title should also be more explanatory.
Figure 2: It is recommended to improve the resolution and increase the size, especially section (c). You could also crop the branches to prioritize CDS, UTRs, introns, and motifs in (a, b) respectively.
Figure 4: This figure is very relevant; however, please consider the following comments: 1) improve the quality and size. 2) your work is with invertebrates, therefore, you may want to consider two similar graphs, one for vertebrates and one for invertebrates only. This may make your results more visible, thereby improving the discussion (giving priority to invertebrates).
Figure 5-8: Add an identifier for each graph (a), (b), etc. To observe gene expression under different treatments, consider grouping Figure 6-8 into a single column. Adding three columns to the time intervals 0…72.
Discussion
I believe the Saline-Alkaline Stress effect is a widely discussed topic, with diverse information on physiological and molecular results. However, your discussion is very simple, comparing your results with a couple of species. I consider this an important detail given the results found. Furthermore, I consider generating subtopics to improve the focus of your discussion, based on your results.
Finally, in L80-82, you justify that your work is related to aquaculture aspects, as well as environmental variables. However, I do not identify a section in the discussion where this is explained. How do your results have implications for aquaculture? Discuss.
L329: Review the entire document; there are repeated omissions.
L338-340: This idea appears to be unfinished.
L358: I consider there are several details in the experimental design:
- The experimental design is unclear; separate the information from the bioassays performed.
- What culture conditions did the control organisms have, N, and what was the N for the analyses?
- What recirculation system was used in each system?
- How were the organisms transported to the laboratory? What welfare measures were used?
- The organisms were placed at a salinity of 22, pH 8.0 ± 0.11, carbonate alkalinity 2.35 ± 0.12 mmol/L, and water temperature of 29 ± 1°C. What was the acclimation time to the new experimental conditions? Was handling stress, etc., taken into account?
- Finally, were the gill samples processed by organism or pool?
L402: add citation
Table 2: add Amplification Efficiency (%), Amplicon Size (bp), Reference
Author Response
Dear editor:
We are truly grateful to the critical comments and thoughtful suggestions from editor and reviewers. Based on these comments and suggestions, we have made careful modifications in the revised manuscript. All changes have been highlighted by yellow color in the revised manuscript. We hope the revised manuscript will meet the standard of International Journal of Molecular Science. The point-by-point responses to the reviewers’ comments were as follows. The original comments are in blue, and our responses are in black.
Reviewer #1
- Title: I recommend adding the common name.
Response: Thank you for this suggestion. We agree with the comment and have added the common name to the title on lines 2-5.
- The abstract sections are well-defined; however, they substantially add to the methodology used (techniques, processes, analysis). Please note the maximum word count.
Response: Thank you for your feedback. We agree and have removed excessive experimental procedures from the abstract, retaining only essential descriptions of results. The abstract now comprises 159 words.
- L43: Negative ions are represented in various ways in the text; apply the same criteria, e.g., L55.
Response: We thank the reviewer for carefully reading our manuscript. We have corrected these on lines 48, 61, 68, 73, 272, and 462.
- L46-50: This section can be more specific, providing examples with other species of aquatic organisms or other aquatic life, as well as mentioning the physiological and molecular changes referred to in more detail. This will allow the reader to better understand the text.
Response: Thank you for your kind advice. We have provided a detailed description of this section and added relevant references on how different salinity and alkalinity levels damage the immune systems and organs in aquatic animals on lines 39-41.
- L62-64: Same as the previous comment, please be more specific.
Response: Thank you for your kind advice. Three concrete examples of saline-stress responses in aquatic animals were added to enhance clarity on lines 50-54 and 69-74.
- L100: Change “,” to “:”
Response: We thank the reviewer for carefully reading our manuscript. We have corrected it on lines 109 and 230.
- Figure 1: The figure is illegible; it is recommended to improve the resolution and increase the size. The title should also be more explanatory.
Response: Thank you for your kind advice. We agree and have improved the resolution and size of Figure 1, provided a high-quality PDF in the supplementary materials, and enhanced its title on lines 112–115.
- Figure 2: It is recommended to improve the resolution and increase the size, especially section (c). You could also crop the branches to prioritize CDS, UTRs, introns, and motifs in (a, b), respectively.
Response: We agree and appreciate your guidance. The size and resolution of Figure 2 were increased to make the CDS, UTR, and motif in (a,b) visible. Figure 2c was modified by removing the extra scale and retaining only one visible scale. A high-quality PDF of Figure 2 was uploaded to the supplementary materials.
- Figure 4: This figure is very relevant; however, please consider the following comments: 1) improve the quality and size. 2) Your work is with invertebrates; therefore, you may want to consider two similar graphs, one for vertebrates and one for invertebrates only. This may make your results more visible, thereby improving the discussion (giving priority to invertebrates).
Response: Thank you for this insightful suggestion. The size and resolution of Figure 4 were improved. However, if two figures were created, the results would be more visual, but this might weaken the comparison of differences in the evolution of vertebrates and invertebrates. We have corrected Figure 4, and the genes from invertebrates are now marked in red font, which improves visibility and makes the comparison results clearer. Meanwhile, detailed explanations were added to the figure captions on lines 166-167.
- Figure 5-8: Add an identifier for each graph (a), (b), etc. To observe gene expression under different treatments, consider grouping Figures 6-8 into a single column. Adding three columns to the time intervals 0…72.
Response: Thank you for your kind advice. We have added identifiers to Figure 5 and Figure 6. We merged Figures 6-8 into one figure as suggested, which makes it easier to observe the expression differences of the same gene under different conditions and helps to select key genes. We added explanations in the figure captions on lines 205-206.
- I believe the Saline-Alkaline Stress effect is a widely discussed topic, with diverse information on physiological and molecular results. However, your discussion is very simple, comparing your results with a couple of species. I consider this an important detail given the results found. Furthermore, I consider generating subtopics to improve the focus of your discussion, based on your results.
Response: Thank you for your kind advice. We have restructured the discussion into subtopics (3.1 and 3.2), added analyses of SLC12 gene deletions in crustaceans on lines 241–254, and revised functional hypotheses for Ecslc12a2.2 on lines 308-320.
- Finally, in L80-82, you justify that your work is related to aquaculture aspects, as well as environmental variables. However, I do not identify a section in the discussion where this is explained. How do your results have implications for aquaculture? Discuss.
Response: Thank you for your kind advice. We have added to this study the importance of aquaculture in the discussion on lines 336-339 and lines 345-347.
- L329: Review the entire document; there are repeated omissions.
Response: We thank the reviewer for carefully reading our manuscript. We have checked the manuscript and removed all duplicated content.
- L338-340: This idea appears to be unfinished.
Response: Thank you for your kind advice. We have corrected the incomplete idea by adding the experimental protocol: Expasy-ProtScale was used to analyze EcSLC12 hydrophobicity on lines 371–372.
- The experimental design is unclear; separate the information from the bioassays performed.
Response: We agree and appreciate this suggestion. The Methods section was reorganized into distinct subsections: Bioinformatics Analysis and Functional Verification on lines 349 and 387-388.
- What culture conditions did the control organisms have, N, and what was the N for the analyses?
Response: Thank you for your kind advice. We have added these on lines 410-412.
- What recirculation system was used in each system?
Response: Thanks for pointing that out. We have added these on lines 398-399.
- How were the organisms transported to the laboratory? What welfare measures were used?
Response: Thank you for your concern for animal welfare. The experiment was conducted at the farm, from selecting shrimp for temporary rearing to tissue collection. The samples were placed in a -80°C freezer after being transported to the laboratory. Therefore, the live shrimp did not need to be transported to the laboratory.
- The organisms were placed at a salinity of 22, pH 8.0 ± 0.11, carbonate alkalinity 2.35 ± 0.12 mmol/L, and water temperature of 29 ± 1°C. What was the acclimation time to the new experimental conditions? Was handling stress, etc., taken into account?
Response: We agree that clarification is needed. This experiment is a 72-hour acute stress experiment. After being selected, the shrimp must be immediately placed under stressful conditions for the experiment.
- Finally, were the gill samples processed by organism or pool
Response: Thank you for this question. Gill tissues from three shrimps per group/time point were pooled into one collection tube for RNA extraction. We have added these on line 413.
- L402: add citation
Response: We agree and have added the requested citation on line 441.
- Table 2: add Amplification Efficiency (%), Amplicon Size (bp), Reference
Response: Thank you for your kind advice. We have added these to Table 2.

Reviewer 2 Report
Comments and Suggestions for Authors
Overall I think the data in the paper are OK and this is a worthwhile study. However, the englsih is poor particularly in the abstract and introduction. There was also an absence of certain information, that while in the methods, could do with being added to the results to give them context and allw better understanding. There are also a couple of things you seem to have gotten wrong or misintepreted but those are easily fixed. here are my specific comments to help you improve the document.
Also I would like to see the authors comment (in the discussion) of the possibility that there may be splice variants of any of their 6 genes and that could complicate the expression analysis and have implications for the functional role(s) of the genes.
Abstract
Line 22 To avoid confusion change ‘EcSLC12’ to ‘The EcSLC12 family'
Line 24 Change ‘was closely related to’ to ‘expression was closely correlated with’
Line 26 Change ‘activated’ to ‘elevated’ change ‘and subsequently’ to ‘and was then subsequently’
Lines 28-29 Change ‘adaption’ to ‘acclimation’
Introduction
Lines 37-41. Just to be clear you are talking about otherwise nominally freshwater, or is this high salinity as its costal/tidal/estuarine areas? You need to clarify what you are talking about here a bit…
Line 41-45 that is a sweeping generalization that is almost certainly not true for all marine animals. I would suggest changing this to (line 42) ‘marine animals that can tolerate such an environment.... Or something similar.
Line 56 somewhere here you need to define AA-permease…
Line 69 I have no idea what is meant by ‘2 salinity’ Do you mean 2 times salinity of seawater? Whatever, you need to better define this…
Results
Line 94. Most likely any sizeable protein would have a hydrophobic index >0 average even if it was a globular non-membrane protein this is due to a greater surface-volume ratio in larger proteins where the inside of globular proteins tends to be net hydrophobic.
Line 130 cite the study then…
Line 136-140 that is a bit of a circular argument given you assigned designations to them based on what they were (homologous to) in the first place.
Line 140-142 What is more interesting is that 2.1 and 2.2 clustered together whereas 2.3 was with another set of species. It suggests the duplication was 2.1/2.2 and 2.3 first and then 2.1 was duplicated to form 2.2 or vice versa.
Line 175-177 here you need to say what they started off in (seems to be seawater in the methods), in terms of environmental water not just what they were acclimated to. It should be written as, from…to…
Also need to say up front (line 175) that this was gill and only gill being measured. Suggest adding it to read ‘analyzing their gill expression’
Line 176 put which figure each set of results was related to here up front, not at the end.
Line 212 need to cite the study if it exists.
Line 229 Say here whether these parameters represent high salinity/alkalinity environment or if not what they do represent.
It would be interesting to know if the numbers for dsRNA-a were significantly different between experiments in figure 9a and 9b. The effect at 48hr and 72 hr seem quite different between the 2 ostensibly same experiments.
Line 221 should be acclimation again not adaptation
Discussion
Line 240 it would be good to say whether these ‘missing’ family members are present in other invertebrates or not (presumably not and they are hence vertebrate specific versions)
Line 242 this is more likely related to the various genome duplication events known to occur in vertebrates that provide scope for functionally divergent gene duplicates
Line 272-274 Suggest citing the original paper for this, which is Cutler and Cramb, 2008 Comp. Physiol. Biochem. 149, 63-73. But probably more relevant in a way would be Cutler and Cramb, 2002, Biochimica Biophysica Acta 1566, 92-103 due to it being about 2 duplicate copies of NKCC1 in teleost fish.
Line 285-287. Not clear what you are referring to here. Presumably a particular NKCC1 protein doesn’t change function just its level of activity? In all this in any case you are assuming that your 3 NKCC1 genes products are still functioning as NKCC1 proteins which you haven’t shown. It would not take too much of a leap for one of them such as SLC12A2.2 to maybe be transporting bicarbonate rather than chloride ions (a bit like the CFTR ion channel can do in vertebrates).
Lines 289-291 I am afraid you have that entirely the wrong way around. NKCC1 is secretory and NKCC2 is absorbative form.
Lines 293-297 First you appear to have the wrong reference as Wu et al [48] appears to be about GAPDH in Western blotting. Second unfortunately the Hiroi study [49] used the T4 antibody for the supposed NKCC1 localizations. That antibody is known to bind to NKCC1, NKCC2 or NCC. Several other more recent studies show the absorbative mitochondrial-rich cells in teleost gill express NCC (probably NCC2 although zebrafish have more types).
Line 304-306. What is the level of potassium in the water. If that is low it would reduce the effectiveness of NKCC for ion absorption (its why teleosts use NCC). I wouldn’t be surprised if SLC12A2.2 has lost its potassium transporting function (I.e. has evolved to be a NCC type) but who knows.
Lines 307-309 Generally speaking you either have NKCC/NCC on one membrane or the other, otherwise they may end up working against each other. You would have Na,KATPase on the basolateral membrane even in invertebrates coupled with a chloride ion channel of some kind (CLC or anoctamin are two possibilities), although permutations using ion exchangers are not impossible.
Line 316 again in your case its acclimation not adaptation (Your shimp may be evolutionarily adapted to the environment they live in but when you change their environment to study their response, you (and they) are acclimating them to the new environment).
Line 425-428 I would change this if I was you as its most likely incorrect. More than likely SLC12A2.1 and SLC12A2.2 are expressed in different cell types in the gill as occurs in teleost fish (NKCCC1 and NCC in that case); see my comment for lines 307-309.
Comments on the Quality of English LanguageEnglish was quite poor in the abstract and in some places in the introduction so I made suggestions for changes that would improve this.
Author Response
Dear editor:
We are truly grateful to the critical comments and thoughtful suggestions from editor and reviewers. Based on these comments and suggestions, we have made careful modifications in the revised manuscript. All changes have been highlighted by yellow color in the revised manuscript. We hope the revised manuscript will meet the standard of International Journal of Molecular Science. The point-by-point responses to the reviewers’ comments were as follows. The original comments are in blue, and our responses are in black.
Reviewer #2
- Line 22 To avoid confusion change ‘EcSLC12’ to ‘The EcSLC12 family'
Response: Thank you for your kind advice. We have updated "EcSLC12" to "The EcSLC12 family" for clarity. This change appears on line 22.
- Line 24 Change ‘was closely related to’ to ‘expression was closely correlated with’
Response: Thank you for your kind advice. The phrasing "was closely related to" has been revised to "expression was closely correlated with" as recommended on line 24.
- Line 26 Change ‘activated’ to ‘elevated’ change ‘and subsequently’ to ‘and was then subsequently’
Response: We thank the reviewer for carefully reading our manuscript. The sentence in question was removed during final editing to streamline the summary.
- Lines 28-29 Change ‘adaptation’ to ‘acclimation’
Response: We thank the reviewer for carefully reading our manuscript. The "Adaptation" has been replaced with "acclimation" throughout the manuscript where appropriate. See line 26.
- Lines 37-41. Just to be clear you are talking about otherwise nominally freshwater, or is this high salinity as its costal/tidal/estuarine areas? You need to clarify what you are talking about here a bit…
Response: Thank you for your kind advice. We have added a definition of saline-alkaline water (distinguishing it from freshwater/marine systems) on lines 36–37.
- Line 41-45 that is a sweeping generalization that is almost certainly not true for all marine animals. I would suggest changing this to (line 42) ‘marine animals that can tolerate such an environment.... Or something similar.
Response: We regret the overgeneralization. The text now specifies "Aquatic animals that can tolerate such an environment" rather than implying all marine animals on lines 44-45.
- Line 56, somewhere here you need to define AA-permease…
Response: Thank you for your kind advice. Definitions for "AA_permease" and "SLC12A domain" have been added on line 63.
- Line 69, I have no idea what is meant by ‘2 salinity’. Do you mean 2 times the salinity of seawater? Whatever, you need to better define this…
Response: We apologize for the confusion. "2 salinity" has been clarified to "salinity 2 ppt" (parts per thousand) on line 78.
- Line 94. Most likely any sizeable protein would have a hydrophobic index >0 average even if it was a globular non-membrane protein this is due to a greater surface-volume ratio in larger proteins where the inside of globular proteins tends to be net hydrophobic.
Response: Thank you for this important insight. We have revised the statement regarding hydrophobicity indices on lines 102-103.
- Line 130 cite the study then…
Response: We have added the requested citation on line 140.
- Line 136-140 that is a bit of a circular argument given you assigned designations to them based on what they were (homologous to) in the first place.
Response: We acknowledge the circular reasoning and have deleted the relevant section.
- Line 140-142 What is more interesting is that 2.1 and 2.2 clustered together whereas 2.3 was with another set of species. It suggests the duplication was 2.1/2.2 and 2.3 first and then 2.1 was duplicated to form 2.2 or vice versa.
Response: We agree with your interpretation. A discussion of the duplication events (2.1/2.2 vs. 2.3) has been added on lines 146–150.
- Line 175-177 here you need to say what they started off in (seems to be seawater in the methods), in terms of environmental water not just what they were acclimated to. It should be written as, from…to…
Response: Thank you for this suggestion. We clarified that shrimp originated in seawater before transfer to test conditions on lines 188.
- Also need to say up front (line 175) that this was gill and only gill being measured. Suggest adding it to read ‘analyzing their gill expression’
Response: We have specified that gene expression was measured in gills on line 186.
- Line 176 put which figure each set of results was related to here up front, not at the end.
Response: We appreciate this guidance. Figure references are now stated upfront in the relevant section on line 187.
- Line 212 need to cite the study if it exists.
Response: We apologize for the oversight. The text now clarifies that this was a pilot study for the current work (revised on line 210).
- Line 229 Say here whether these parameters represent high salinity/alkalinity environment or if not what they do represent.
Response: Parameters now explicitly describe a saline-alkaline environment on lines 226-227, 228–229.
- It would be interesting to know if the numbers for dsRNA-a were significantly different between experiments in Figure 9a and 9b. The effect at 48h and 72 h seem quite different between the 2 ostensibly same experiments.
Response: Thank you for noting this discrepancy. Figures 9a and 9b were conducted under different experimental conditions. Figure 9a was conducted under seawater conditions to select the target with the most significant interference effect. Figure 9b was conducted in saline-alkaline water. This may be the primary reason for the difference in dsRNA-a values between the two. To prevent reader misunderstanding, we have provided clarification in the figure caption (lines 224-225) and the experimental methods section (lines 424-426) of the manuscript.
- Line 221 should be acclimation again, not adaptation
Response: We have replaced "adaptation" with "acclimation" as suggested on line 222.
- Line 240 it would be good to say whether these ‘missing’ family members are present in other invertebrates or not (presumably not and they are hence vertebrate specific versions)
Response: Thank you for your kind advice. Discussion of missing SLC12 family members in invertebrates (vs. vertebrates) was added on lines 241–254.
- Line 242 this is more likely related to the various genome duplication events known to occur in vertebrates that provide scope for functionally divergent gene duplicates
Response: Thank you for your kind advice. The large number of members in the vertebrate SLC12 gene family is likely related to genome duplication. We have revised the previous conclusion on lines 255-265.
- Line 272-274 Suggest citing the original paper for this, which is Cutler and Cramb, 2008 Comp. Physiol. Biochem. 149, 63-73. But probably more relevant in a way would be Cutler and Cramb, 2002, Biochimica Biophysica Acta 1566, 92-103 due to it being about 2 duplicate copies of NKCC1 in teleost fish.
Response: Thank you for the key references. We have cited Cutler & Cramb (2002, 2008) on lines 282–286.
- Line 285-287. Not clear what you are referring to here. Presumably a particular NKCC1 protein doesn’t change function just its level of activity? In all this in any case you are assuming that your 3 NKCC1 genes products are still functioning as NKCC1 proteins which you haven’t shown. It would not take too much of a leap for one of them such as SLC12A2.2 to maybe be transporting bicarbonate rather than chloride ions (a bit like the CFTR ion channel can do in vertebrates).
Response: We apologize for the lack of clarity. The text now explicitly contrasts EcSLC12A2.2’s low-salinity upregulation with typical NKCC1 high-salinity responses and notes its potential functional divergence (e.g., bicarbonate transport) on lines 298–306.
- Lines 289-291 I am afraid you have that entirely the wrong way around. NKCC1 is secretory and NKCC2 is absorbative form.
Response: We regret the error. NKCC1 (secretory) and NKCC2 (absorptive) functions are now correctly described on lines 300–301.
- Lines 293-297 First you appear to have the wrong reference as Wu et al [48] appears to be about GAPDH in Western blotting. Second unfortunately the Hiroi study [49] used the T4 antibody for the supposed NKCC1 localizations. That antibody is known to bind to NKCC1, NKCC2 or NCC. Several other more recent studies show the absorbative mitochondrial-rich cells in teleost gill express NCC (probably NCC2 although zebrafish have more types).
Response: We have replaced the incorrect citation with the appropriate reference on lines 635–636.
- Line 304-306. What is the level of potassium in the water. If that is low it would reduce the effectiveness of NKCC for ion absorption (its why teleosts use NCC). I wouldn’t be surprised if SLC12A2.2 has lost its potassium transporting function (I.e. has evolved to be a NCC type) but who knows.
We appreciate the reviewer’s insightful comment. After reviewing the latest literature, we agree that EcSLC12A2.2 is likely to have evolved NCC-like functions. Accordingly, we have addressed this point in lines 308–320 of the revised manuscript. However, we did not measure potassium ion concentrations in the experimental water, which limits our ability to definitively classify EcSLC12A2.2 as an NCC-type transporter. This limitation highlights an important direction for future investigation.
- Lines 307-309 Generally speaking, you either have NKCC/NCC on one membrane or the other; otherwise, they may end up working against each other. You would have Na,KATPase on the basolateral membrane even in invertebrates, coupled with a chloride ion channel of some kind (CLC or anoctamin are two possibilities), although permutations using ion exchangers are not impossible.
Response: We completely agree with your point of view. EcSLC12A2.1 and EcSLC12A2.2 are now hypothesized to operate in distinct ionocytes (like teleost models), avoiding functional conflict. Revised on lines 321-322.
- Line 316 again in your case its acclimation not adaptation (Your shimp may be evolutionarily adapted to the environment they live in but when you change their environment to study their response, you (and they) are acclimating them to the new environment).
Response: We appreciate this critical distinction. "Adaptation" has been replaced with "acclimation" on line 324.
- Line 425-428 I would change this if I was you as its most likely incorrect. More than likely SLC12A2.1 and SLC12A2.2 are expressed in different cell types in the gill as occurs in teleost fish (NKCCC1 and NCC in that case); see my comment for lines 307-309.
Response: We agree with your interpretation. The conclusion now aligns with teleost models: EcSLC12A2.1 and A2.2 likely localize to different gill cell types. Revised on lines 321–322 and 461–462.

Round 2
Reviewer 1 Report
Comments and Suggestions for Authors
In this second revision, a considerable improvement in the work can be seen.
I consider the work suitable for publication.
Author Response
- The English could be improved to more clearly express the research
Response: Thank you for your time and constructive comments on our manuscript. We have thoroughly polished the entire manuscript to improve the clarity, flow, and accuracy of the English language. We have paid special attention to simplifying complex sentences, correcting grammatical errors, and employing more precise academic terminology to ensure our research is presented as clearly as possible.
Reviewer 2 Report
Comments and Suggestions for Authors
A much improved document but I still see 3 things that were not quite right yet. See comments below...
Line 78 Change to read ‘ at a salinity of 2 ppt’
Line 148-150 Still does not read right to my mind. It should be “This suggests that the duplication event giving rise to the branches containing EcSLC12A2.1/2.2 and EcSLC12A2.3 occurred earlier and this was then followed by a further duplication that produced the EcSLC12A2.1 and EcSLC12A2.2 genes.”
Line 285 Although it would be unsurprising to have alternative splicing in these genes, the NKCC1a/b, NKCC2α/β are both pairs of separate genes (e.g. NKCC1a and NKCC1b or NKCC2α and NKCC2β) with completely unique sequences and are not generated by alternative splicing but by gene duplications. Please amend the text to reflect this.
Author Response
Dear editor:
We are truly grateful to the critical comments and thoughtful suggestions from editor and reviewers. Based on these comments and suggestions, we have made careful modifications in the revised manuscript. All changes have been highlighted by yellow color in the revised manuscript. We hope the revised manuscript will meet the standard of International Journal of Molecular Science. The point-by-point responses to the reviewers’ comments were as follows. The original comments are in blue, and our responses are in black.
- Line 78 Change to read ‘at a salinity of 2 ppt’
Response: We agree with this comment. The text has been revised to "at a salinity of 2 ppt" as recommended at lines 78-79.
- Line 148-150 Still does not read right to my mind. It should be “This suggests that the duplication event giving rise to the branches containing EcSLC12A2.1/2.2 and EcSLC12A2.3 occurred earlier and this was then followed by a further duplication that produced the EcSLC12A2.1 and EcSLC12A2.2 genes.”
Response: Thank you for your kind advice. We made corrections to lines 149-152 of the manuscript.
- Line 285 Although it would be unsurprising to have alternative splicing in these genes, the NKCC1a/b, NKCC2α/β are both pairs of separate genes (e.g. NKCC1a and NKCC1b or NKCC2α and NKCC2β) with completely unique sequences and are not generated by alternative splicing but by gene duplications. Please amend the text to reflect this.
Response: We fully agree with this correction. The text in lines 283–289 has been revised to clearly distinguish gene duplication events from alternative splicing mechanisms.
